# Variability in body weight and morphology of Uganda's indigenous goat breeds across agroecological zones

Ziwena Nantongo[1,2,3]*, Morris Agaba[2], Gabriel Shirima[2‡], Swidiq Mugerwa[3], Stephen Opiyo[4], Raphael Mrode[1], Josephine Birungi[1], Linus Munishi[2‡]

**1** Biosciences Eastern and Central Africa, International Livestock Research Institute, Consortium of International Agricultural Research Centers (CGIAR), Nairobi, Kenya, **2** School of Life Sciences, Nelson Mandela African Institution of Science and Technology, Arusha, Tanzania, **3** National Livestock Resources Research Institute, National Agricultural Research Organization, Kampala, Uganda, **4** Molecular and Cellular Imaging Center, The Ohio State University, Columbus, Ohio, United States of America

☯ These authors contributed equally to this work.
‡ GS and LM also contributed equally to this work.
* nantongozuena@gmail.com

**Data Availability Statement:** All relevant data are within the manuscript and its Supporting information files.

## Abstract

Indigenous goat breeds in Uganda are classified based on average body size parameters and coat color. However, variations in the body size of animals may be influenced by several factors, including management and the environment. To understand the effect of the agro-ecological zone on the physical characteristics and live weight of Uganda's indigenous goats, this study evaluated the body size characteristics of the three indigenous goat breeds of Uganda across ten agroecological zones. The cross-sectional survey was conducted in 323 households from the ten zones, where 1020 goats composed of three breeds (Mubende, Kigezi, and Small East African) were sampled and measured for body weight, linear body size, and age. We confirmed that Mubende and Kigezi goats from the original homeland had a higher mean body weight than reported in FAO reports. In addition, Mubende appeared to perform better in pastoral rangelands, with a higher mean body weight (38.1 kg) and body size being significantly higher (P < 0.0001) compared to other zones. The mean body weight for the Kigezi breed in the original homeland (34 kg) was comparable to those from Western Savannah grasslands and pastoral rangelands and less than that initially reported by FAO (30 kg). Similarly, there was no significant difference in the linear body size characteristics of Kigezi goats in the home zone of highland ranges relative to those found in other agroecological zones (P > 0.05). Although the Small East African goats were originally found in Northwestern Savannah grassland and Northeastern dryland zones, they performed poorly regarding mean body weight and body size characteristics in the former zone. In the Northwestern Savannah grasslands, the mean body weight (23.8 kg) was even less than that reported by FAO, which ranged between 25 and 30 kg. Finally, we confirmed that Mubende and Kigezi goats are significantly heavier than small East African goats (p ≤ 0.0001). The results of this study can be useful in designing precise management strategies to improve indigenous goat productivity in different environments in Uganda.

**Funding:** The research was funded by Swedish International Development Cooperation Agency (Sida) through a grant to Biosciences eastern and central Africa—International Livestock Research Institute (BecA-ILRI Hub) (Sida contribution no: 51050080) to NZ as a PhD fellow under the Africa Biosciences Challenge Fund (ABCF) program. The ABCF Program is funded by the Australian Department for Foreign Affairs and Trade (DFAT) through the BecA-CSIRO partnership, the Syngenta Foundation for Sustainable Agriculture (SFSA), the Bill & Melinda Gates Foundation (BMGF), the UK Department for International Develop-ment (DFID), and the Swedish International Development Cooperation Agency (Sida).

**Competing interests:** The authors have declared that no competing interests exist.

## 1. Introduction

Agriculture is among the crucial sectors in Uganda's economy, contributing 23.84% to the country's gross domestic product (GDP) and 72.4% to employment [1]. Livestock contributes 4% to the total GDP and 16% to the agricultural GDP through the production of cattle, goats, poultry, pigs, and sheep [2]. Goats are an important component of Uganda's livestock sector, especially in resource-poor communities where they serve diverse roles. They are a domain in the traditional wealth transformation series that is convertible to cattle and subsistence cash for household needs [3]. Besides income, goats provide meat and milk and play socio-cultural roles like dowry and prestige.

Uganda's goat population is 10.4 million [4], 93.4% of which are indigenous [5], and small-holder farmers mainly own them because their management requires less input and goats are easy to dispose of. Indigenous goats in Uganda, as described by the Food and Agricultural Organization (FAO), comprise three breeds, namely Mubende, Kigezi, and Small East African [6], which exist in proportions of 35.6%, 11.2% and 53.2%, respectively, relative to the total population [7]. Mubende goats derive their name from their home ground, Mubende district, in the current Western savannah grasslands zone, but also live in areas towards the south and around Lake Victoria [6]. They are primarily black or black and white with a short, fine, shiny hair coat. Female Mubende goats weigh 31kg of live body weight, while their male counter-parts weigh 35 kg. The Kigezi goats, from the former Kigezi district in southwest Uganda, which is in the current highland ranges agroecological zone, are known for their long, curly hair, especially on the hind quarters. Their average live body weight is 30 kg. Small East Afri-can (SEA) goats are known to exist in all regions of Uganda, especially in the northwestern savannah grasslands and northeastern drylands [8]. SEA goats are small with a live body weight between 25 and 30 kg and have a fine, short hair coat with varying colors [8]. Uganda's indigenous goats are a loosely characterized animal resource whose sustainability is threatened by indiscriminate crossbreeding and climate change effects. Although the National Animal Genetic Resource Center and Data Bank strive to conserve Uganda's indigenous goats in situ, insufficient information on distinct breed identifiers is challenging. Indigenous goat character-ization is further complicated by a lack of farm records [9], especially on pedigree and goat production, yet haphazard management threatens biodiversity conservation. Furthermore, the introduction of exotic breeds like Toggenburg, Anglonubian, Saneen, Alpine, and Boer, to mate with the indigenous goats for improvement of production-related traits [7] may lead to the loss of the yet-to-be-characterized diversity of indigenous goats. However, the characteri-zation of animal breeds and populations, including their genetic differentiation and relation-ships, is widely recognized for conservation and utilization efficiency. It is, therefore, important that the traits of indigenous goats in Uganda are documented, especially within the different environments where they survive.

Indigenous goats are distributed in all regions of Uganda despite variations in environmen-tal characteristics. Uganda is divided into ten agroecological zones based on climatic, vegeta-tion, and altitude variations [10]. Generally, the mountainous areas are cool and moist; central to southwestern areas are warm and humid; areas close to water bodies are hot and humid; and areas in the north, towards eastern Uganda, are hot and dry. The livelihood of people in drier zones is dominated by extensive, low-input livestock production, with goats ranked as second most important after cattle. In contrast, people in the warm-humid zones practice a mixed-crop-livestock production system. However, global warming trends have been observed in different regions of Uganda, with variations in warming rates from 1981 to 2010 [11]. The western and southwestern regions experienced the highest warming rate at 1.49˚ C per decade, followed by the central, eastern, and northern regions at 1.17˚ C, 0.67˚ C, 0.6˚ C, respectively.

Therefore, climate change adaptation mechanisms are needed for a sustainable livestock sector, including goat production. The conservation of Uganda's environmentally adapted indigenous goat genetic resources will assure goat breeders and farmers of the necessary flexibility in climate change adaptation [12]. However, the shortage of information on the extent of diversity in indigenous goats across agroecological zones of Uganda limits informed decision-making for the goats' sustainable utilization and conservation.

Documenting phenotypic characteristics is the first step towards developing strategies for sustainable use, development, and conservation of genetic resources [13]. Therefore, this study intended to evaluate the phenotypic characteristics of the three Ugandan goat breeds with reference to FAO characterization in their original homeland agroecological zones [6] and understand their current status across different agroecological zones. In addition, linear morphometric measurements were assessed for each goat breed to provide additional information on existing phenotypic descriptions of Uganda's indigenous goat breeds.

## 2. Materials and methods

### 2.1 Ethical statement

This study is part of an extensive study entitled "Population Structure, Genetic Diversity, and Selection Signatures in Indigenous Goat Breeds from Different Agroecological Zones of Uganda." The study was approved by the International Livestock Research Institute's (ILRI) Institutional Research Ethics Committee (IREC) (Protocol code ILRI-IREC2019-19 and date of approval 1st July 2019). The animal study protocol was approved by the International Livestock Research Institute's (ILRI) Institutional Animal Care and Use Committee (IACUC) (Protocol code IACUC 2019–19 and date of approval 24th July 2019). A written consent statement was read to each participant, and a yes-or-no response was recorded for each questionnaire before collecting the data. All study participants were also the owners of the animals involved in the study, and only those who consented to participation and the use of their animals for the study were involved.

### 2.2 Study area

This study covered Uganda, encompassing all its ten agroecological zones. Uganda is located in the East Africa region, bordered on the east by Kenya, on the west by the Democratic Republic of the Congo (DRC), on the north by Sudan, on the southwest by Rwanda, and the south by Tanzania. Uganda's climate is tropical, though temperatures cool with increasing altitude. Its annual rainfall ranges from more than 2,100 millimeters around Lake Victoria to about 500 millimeters in the northeast. Based on climate, vegetation cover, and altitude differences, Uganda is stratified into ten agroecological zones: northeastern drylands, northeastern savannah grasslands, northwestern savannah grasslands, para-savannahs, Kyoga plains, Lake Victoria crescent, western savannah grasslands, pastoral rangelands, southwestern farmlands, and the highland ranges. Goat production occurs in all agroecological zones dominated by indigenous breeds (Mubende, Kigezi, and Small East African), but each breed has a known home zone. Mubende goats originate from the western savannah grasslands zone, characterized by warm and humid temperatures, in a rainforest-rich natural savanna grassland vegetation [10]. The zone farming system is described as a banana-coffee-cattle system, where cattle are the primary livestock along with small ruminants [3]. Livestock production is mainly extensive management, with grazing in open, non-cultivable areas and fenced-off farms. Supplementation with crop residues after cropping seasons is a common practice among farmers. Mubende goats are mainly black or black and white with a short, shiny hair coat, as

described by FAO [6, 8]. Also, the mean body weight of female Mubende goats, according to FAO from the homeland, was 31 kg (Fig 1A).

Kigezi goats originate from the highland ranges zone, an area characterized by a cool and moist environment where vegetation is a mixture of high-altitude forest, savannah mosaic at high altitudes, montane forest, and high open moorland [10]. The zone is characterized by high human population density with small land holdings, and farmers keep small ruminants, cattle, and poultry alongside sorghum, potatoes, vegetables, coffee, maize, and wheat [3]. Goats either graze in open, non-cultivatable areas or are tethered. Kigezi goats are known for their long, curly hair coat, especially on the hindquarters [6, 8] (Fig 1B).

Small East African goats originate from northwestern savannah grassland, an area characterized by a hot and humid climate in savanna vegetation with open mixtures of trees and shrubs standing within tall grass [10]. The northeastern drylands zone is also known as a home zone for small East African goats. It is a hot and dry environment where vegetation has thorny bushes, cammiphora woodlands, occasional small trees, and patches of grassland. Small East African goats are small bodied goats and appear in multiple colors [8] (Fig 1C).

## 2.3 Study design and sampling approach

The study followed a cross-sectional design in all agroecological zones [14] where all selected farmers were visited. Written consent was read to each participant for approval before the study, and only those who consented to the study participated. A multi-stage systematic random sampling technique was used, starting at the country level, where all the ten zones were considered for the selection of the farms and individual animals, which was based on study selection criteria [15]. Within each agroecological zone, two districts were randomly selected, and 20 farms per district were selected based on the study inclusion/exclusion criteria. Selected farms had a flock of at least 10 indigenous goats that included at least 3 does of age 2 or more years and a location of about 10 kilometers away from the previously recruited farm. Farms that did not meet all three criteria were excluded from the study. At the farms, goats were segregated into the known indigenous breeds according to FAO classification as Mubende, Kigezi, and Small East African, based on hair coat type and hair coat color [6, 8] (Fig 1). After separation into the identified breeds, three mature female goats of age ≥ 2 years, having different family lines, were selected within each breed. Three goats were selected per breed as replicates per farm per breed to capture any farm level diversity which contributed to the total diversity for the zone. Farm records and/or farmer's memory of goat kinship were used to ensure selected goats were from different family lines. In cases where a farm had more than one indigenous breed but less than three qualifying mature female goats for selection per breed, only qualifying female goats in each breed were selected. If all mature goats at the farm were from one family line, only one was recruited into the study; this scenario was

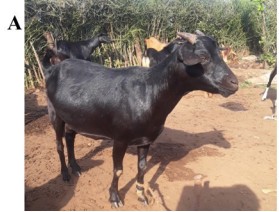
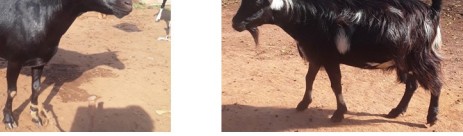
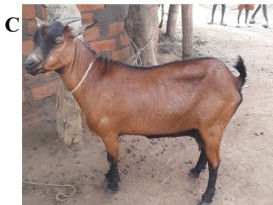

**Mubende goat**    **Kigezi goat**    **Small East African goat**

**Fig 1. Pictorial of the different indigenous goat breeds in Uganda as described by FAO (6).** Source: Nantongo Ziwena.

encountered in 3 of the households that participated. For each selected goat, body-size characteristics were individually measured and recorded. All data collected at farm level in each zone was put together as total dataset for each zone.

## 2.4 Data collection

Data collection started in northeastern through the mid-north to northwestern and western Uganda between July and October 2019. More data were collected in southwestern, southern, and central Uganda between October and December 2020 to cover all the 10 agroecological zones (Fig 2). At each farm, the data collected included flock characteristics such as the total number of goats, the number of does, bucks and kids. In addition, linear morphological traits were measured on each selected goat using graduated tapes following the recommended FAO guidelines [13] as in [15] (Fig 3). The traits measured were: body length (BL), withers height (WH), rump height (RH), and heart girth (HG). Body weight (BW) was measured using a portable weighing scale (Hanson Heavy duty portable hook type 100 Kg weighing scale with ±10 gm error), where a goat was carefully laid onto a nylon sack secured with strings at the four

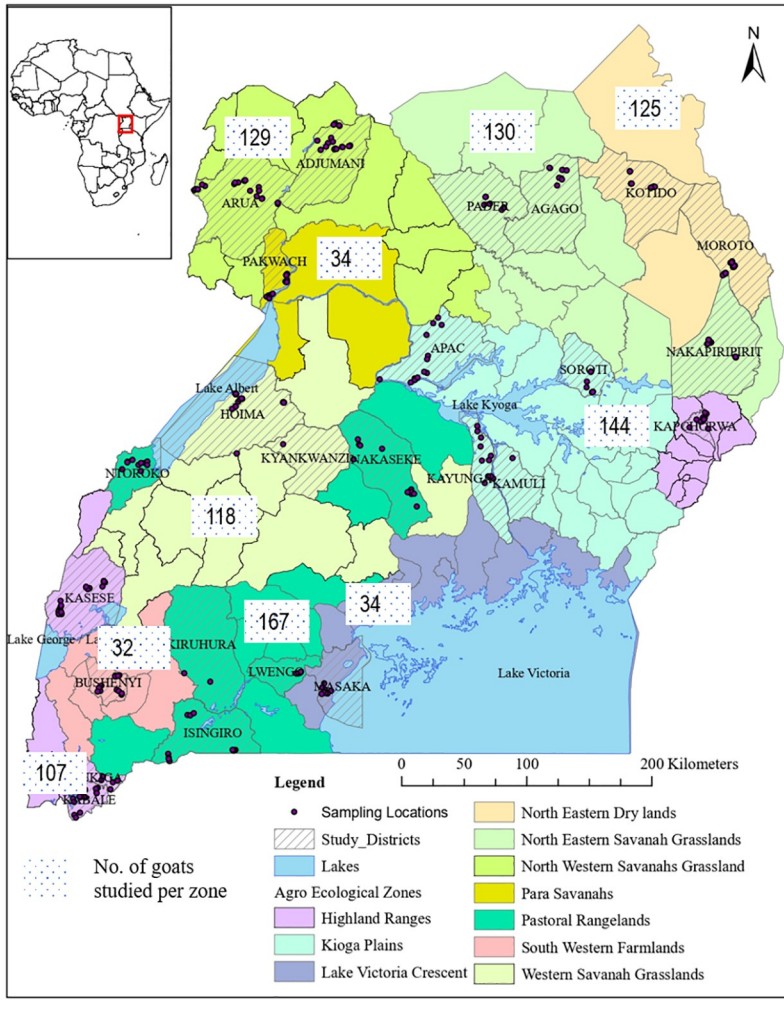

**Fig 2. A map of Uganda showing study sites, and the number of goats selected per agroecological zone.** Source: Humanitarian Data Exchange and Nantongo Ziwena.

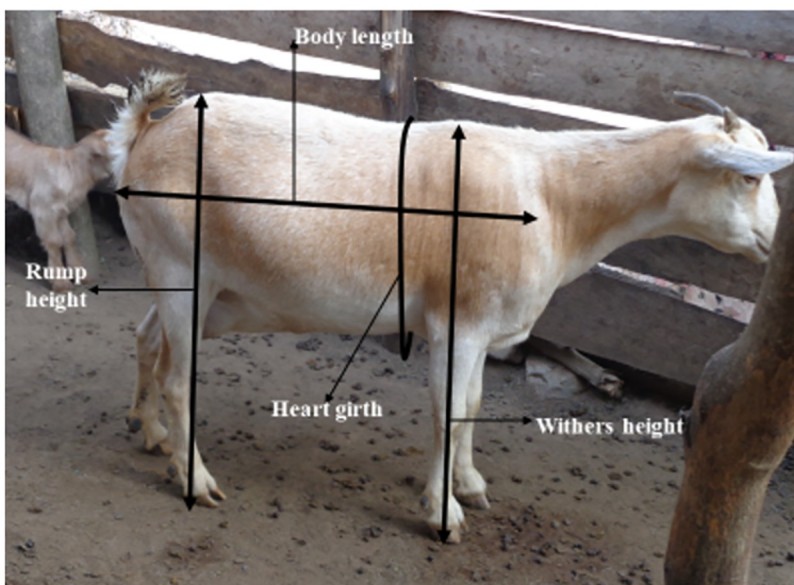

**Fig 3. Illustration of the different body size measurements taken.** Source: Nantongo Ziwena.

corners. The nylon bag with the goat was carefully hung onto the balance hook by the strings, and the reading was quickly taken. The age of each goat was estimated using the dentition method following the guidelines in [16]. Data were recorded using a structured questionnaire uploaded on the Open Data Kit (ODK) and directly saved to a server at the International Livestock Research Institute (ILRI-Nairobi).

## 2.5 Data analysis

Data were downloaded from the server in Excel format, cleaned for errors by filtering for outliers and tested for normality by comparing a histogram of the dataset to the normal probability curve.

**2.5.1 Analysis of farm characteristics.** Frequency distributions, range and median for total number of does, bucks and kids per farm across agroecological zones were generated using Microsoft Excel spreadsheet software (Microsoft 365) to understand flock size characteristics.

**2.5.2 Analysis of physical characteristics.** Using R software, version 4.1.1 (2021-08-10)–"Kick Things" package, body weight, and linear body measurements of goats were analyzed for each breed across agroecological zones. Age was clustered into three categories (age ranges): 2 to 3, > 3 to 4, and above 4 years. Descriptive statistics were estimated for each breed across agroecological zones and age ranges. Analysis of covariance (ANCOVA) was used to compare means of body weight and linear body size characteristics with age as a covariant for each goat breed in the home zone and across other agroecological zones where they were found, at 95% confidence interval, and significant differences in mean values were separated by Tukey HSD post hoc test. The ANCOVA model used was

$$Y_{ij} = \mu + \alpha_i + \beta x_{ij} + \varepsilon_{ij},$$

where
$Y_{ij}$ is the observed value of body weight/body size parameter of a goat within a zone

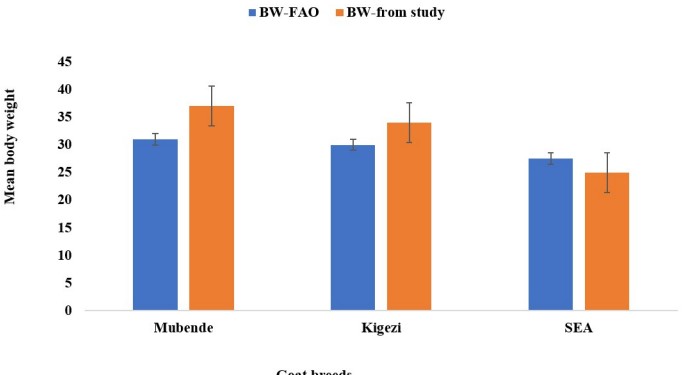

**Fig 4. Overall mean body weight with standard error for each goat breed studied versus the FAO record(6).**

μ is the overall mean
$\alpha_i$ is the effect of the agroecological zone
β is the effect of age as a covariate
$x_{ij}$ is the value of age of a goat within a zone
$\varepsilon_{ij}$ is the residual error

## 3. Results

### 3.1 Goat farm characteristics

A total of 1020 goats were studied, of which 525 were Mubende, 430 were Small East African and 65 were Kigezi goats. The total goats per household ranged from 10 to 490, with a median of 24 goats. The flocks were composed of 3 to 420 does with a median of 13 does, 0 to 50 bucks with a median of 3 bucks, and 0 to 102 kids with a median of 7 kids. Overall, the flock structure comprised 62.7% does, 11.8% bucks, and 25.6% kids. Goat management on 77.7% of the farms involved extensive grazing, 6.6% practiced grazing with some crop residue supplementation, 14.1% tethering, and 1.4% cut and carry system.

### 3.2 Mean body weight for each breed in the zone of origin against the FAO record

The current measurements revealed that the overall mean body weights of the Mubende and Kigezi breeds were higher compared to FAO data, while that of the Small East African breed was lower (Fig 4).

### 3.3 Physical characteristics of goat breeds within their current homeland and across different agroecological zones

To evaluate differences in body size of each breed in the known zone of origin relative to other zones where it is was found, bodyweight and linear body measurements of each breed were compared across agroecological zones after removing the confounding effect of age. The mean weight of Mubende goats from their home zone of Western savannah grasslands was 36.69kg compared to 38.11kg found in the Pastoral rangelands (Table 1). However, linear body size measurements were significantly (p < 0.0001) higher for Mubende goats in pastoral range-lands than in western savannah grasslands. Mubende goats in other agroecological zones had significantly (p < 0.0001) lower body weight and linear body size characteristics compared to

**Table 1. Mean body size of Mubende goats from different agro-ecological zones.**

| Zone (n) | Body weight | Body length | Chest girth | Withers height | Rump height |
|---|---|---|---|---|---|
| HLR (70) | 32.94 ± 0.275[b] | 62.88 ± 0.188[b] | 74.04 ± 0.226[c] | 61.08 ± 0.134[ab] | 62.38 ± 0.136[a] |
| KP (76) | 32.49 ± 0.370[b] | 62.46 ± 0.181[b] | 73.53 ± 0.263[c] | 59.86 ± 0.127[b] | 61.06 ± 0.120[b] |
| LVCS (33) | 32.42 ± 0.329[b] | 62.35 ± 0.182[b] | 71.43 ± 0.225[c] | 59.88 ± 0.224[b] | 59.88 ± 0.074[bc] |
| NWSGL (43) | 29.92 ± 0.243[b] | 60.64 ± 0.199[b] | 71.59 ± 0.206[c] | 57.12 ± 0.137[c] | 59.69 ± 0.132[c] |
| PR (138) | 38.11 ± 0.343[a] | 66.67 ± 0.192[a] | 78.30 ± 0.249[b] | 62.46 ± 0.137[a] | 63.28 ± 0.138[a] |
| PS (20) | 31.10 ± 0.208[b] | 60.07 ± 0.170[b] | 72.64 ± 0.182[c] | 58.74 ± 0.118[bc] | 60.26 ± 0.138[b] |
| SWFL (32) | 33.66 ± 0.326[b] | 63.42 ± 0.179[ab] | 74.37 ± 0.217[c] | 61.67 ± 0.147[a] | 62.31 ± 0.132[ab] |
| WSGL (111) | 36.69 ± 0.395[a] | 64.31 ± 0.340[a] | 80.45 ± 0.308[a] | 58.54 ± 0.196[b] | 62.92 ± 0.184[a] |
| Significance | *** | *** | *** | *** | *** |
| Overall (523) | 34.73 ± 0.357 | 63.83 ± 0.244 | 76.05 ± 0.289 | 60.27 ± 0.173 | 62.08 ± 0.152 |

Significance codes: 0 '***' 0.001 '**' 0.01 '*' 0.05 '.' 0.1 ' ' 1

[a-d]Within column values with differing letters are significantly different. Agroecological zones: HLR- Highland ranges; KP- Kyoga plains; LVCS-Lake Victoria crescent; NWSGL- Northwestern savannah grasslands; PR- Pastoral rangelands; PS- Para savannah; SWFL-Southwestern farmlands; WSGL- Western savannah grasslands

those in western savannah and pastoral rangelands. Mubende goats in northwestern savannah grasslands had the lowest body weight and linear body characteristics.

Kigezi goats from the original zone, highland ranges had the highest mean body weight (34.20 kg), which was comparable to the mean weight observed in Western savannah grasslands (33.5 kg) and pastoral rangelands (34.23 kg) (Table 2). Similarly, there was no significant difference in the linear body size characteristics except for height at withers of Kigezi goats in the home zone of highland ranges, which was significantly higher than for the same goats in western savannah grasslands.

Small East African (SEA) goats are known to originate from the northwestern savannah grasslands zone and the northeastern drylands zone. However, Small East African goats found in the northwestern savannah grasslands had the lowest body weight and linear body characteristics compared to those found in the northeastern drylands. (Table 3). In relation to other agroecological zones where the Small East African goats were found, the body weight and linear body size characteristics were significantly comparable to those in the northeastern drylands except for para-savannah and Kyoga plains where the goats had significantly lower body size characteristics. Small East African goats with lowest body size were found in northwestern savannah grasslands, para-savannahs and Kyoga plains.

Overall comparison between breeds showed that the mean body weights of Mubende and Kigezi goats (34.73 ± 0.357 kg and 34.0 ± 0.862 kg) respectively, were significantly higher (p

**Table 2. Mean body size of Kigezi goats from different agro-ecological zones.**

| Zone (n) | Body weight | Body length | Chest girth | Withers height | Rump height |
|---|---|---|---|---|---|
| HLR (14) | 34.20 ± 0.759 | 64.35 ± 0.511 | 73.66 ± 0.00 | 61.81 ± 0.511[a] | 63.50 ± 0.766 |
| PR (22) | 34.23 ± 0.823 | 64.42 ± 0.676 | 75.85 ± 1.103 | 61.60 ± 0.471[ab] | 62.46 ± 0.504 |
| WSGL (8) | 33.38 ± 1.085 | 62.97 ± 0.962 | 76.36 ± 0.864 | 58.26 ± 0.547[b] | 61.91 ± 0.525 |
| Significance | ns | ns | ns | * | ns |
| Overall (44) | 34.0 ± 0.862 | 64.55 ± 0.721 | 75.76 ± 0.988 | 60.81 ± 0.541 | 62.42 ± 0.519 |

Significance codes: 0 '***' 0.001 '**' 0.01 '*' 0.05 '.' 0.1 ' ' 1, ns-not significant

Agroecological zones: HLR- Highland ranges; KP- Kyoga plains; LVCS-Lake Victoria crescent; NWSGL- Northwestern savannah grasslands; PR- Pastoral rangelands; PS- Para savannah; SWFL-Southwestern farmlands; WSGL- Western savannah grasslands

**Table 3. Mean body size of Small East African goats from different agro-ecological zones.**

| Zone (n) | Body weight | Body length | Chest girth | Withers height | Rump height |
|---|---|---|---|---|---|
| HLR (20) | 27.90 ± 0.263[ab] | 59.69 ± 0.204[a] | 71.37 ± 0.256[ab] | 59.56 ± 0.192[b] | 61.34 ± 0.187[a] |
| KP (53) | 25.13 ± 0.491[bc] | 58.16 ± 0.178[ab] | 66.62 ± 0.222[bc] | 56.84 ± 0.143[c] | 58.40 ± 0.118[b] |
| NEDL (119) | 28.34 ± 0.257[a] | 59.48 ± 0.199[a] | 73.00 ± 0.307[a] | 61.77 ± 0.164[a] | 62.05 ± 0.157[a] |
| NESGL (117) | 27.39 ± 0.262[ab] | 59.39 ± 0.218[ab] | 70.99 ± 0.341[ab] | 60.33 ± 0.187[b] | 61.08 ± 0.175[a] |
| NWSGL (82) | 23.90 ± 0.240[c] | 57.72 ± 0.237[ab] | 66.97 ± 0.233[abc] | 54.94 ± 0.151[c] | 57.30 ± 0.176[b] |
| PS (11) | 24.23 ± 0.217[abc] | 57.96 ± 0.146[ab] | 68.12 ± 0.235[abc] | 57.38 ± 0.107[bc] | 59.23 ± 0.116[ab] |
| WSGL (13) | 26.33 ± 0.228[abc] | 67.10 ± 0.392[a] | 74.51 ± 0.219[a] | 55.46 ± 0.036[c] | 60.54 ± 0.130[a] |
| Significance | *** | *** | *** | *** | *** |
| Overall (415) | 26.62 ± 0.305 | 58.96 ± 0.215 | 70.18 ± 0.315 | 59.08 ± 0.209 | 60.23 ± 0.185 |

**Significance codes**: 0 '***' 0.001 '**' 0.01 '*' 0.05 '.' 0.1 ' ' 1

[a-c]Within column values with differing letters are significantly different. **Agro-ecological zones**: HLR- Highland ranges; KP- Kyoga plains; NEDL-Northeastern dry lands; NESGL-Northeastern savannah grass lands; NWSGL- Northwestern savannah grasslands; PS- Para savannah; WSGL- Western savannah grasslands;

≤0.0001) than 26.62 ± 0.305kg for Small East African goats. A similar trend was observed for linear body characteristics, where body length, chest girth, withers height, and rump height for Mubende and Kigezi goats were significantly higher as compared to the same measurements for Small East African goats.

### 3.4 Body size characteristics of goat breeds across age ranges

The body size of goat breeds was evaluated across age ranges after removing the confounding effect of age for each animal. Results showed a general trend of increased body weight and linear morphological characteristics with increasing age range (Table 4). The interaction of breed and age range was significant for body weight and chest girth; thus, within a breed, age category influences the observed body weight and chest girth of a goat, and likewise, within each age category, the goat breed has an influence on its body weight and chest girth. Although there was a non-significant interaction of breed and age range for body length, withers height, and rump height, the main effects of both breed and age category were significant. Therefore,

**Table 4. Means of body size characteristics for different breeds across age ranges.**

| Breed (n) | Age range | Body weight | Body length | Chest girth | Withers height | Rump height |
|---|---|---|---|---|---|---|
| Kigezi (10) | 2 to 3 | 33.05 ± 0.165 | 62.39 ± 0.128 | 75.69 ± 0.119 | 59.94 ± 0.084 | 61.72 ± 0.087 |
| Mubende (153) | 2 to 3 | 31.98 ± 0.234 | 61.67 ± 0.140 | 74.85 ± 0.236 | 58.27 ± 0.106 | 61.32 ± 0.102 |
| Small East African (139) | 2 to 3 | 23.18 ± 0.159 | 56.65 ± 0.135 | 66.55 ± 0.173 | 57.42 ± 0.125 | 58.58 ± 0.101 |
| Kigezi (16) | >3 to 4 | 34.47 ± 0.168 | 64.79 ± 0.121 | 75.01 ± 0.138 | 61.04 ± 0.104 | 62.71 ± 0.086 |
| Mubende (173) | >3 to 4 | 33.77 ± 0.270 | 63.75 ± 0.214 | 74.90 ± 0.248 | 60.39 ± 0.132 | 61.84 ± 0.125 |
| Small East African (129) | >3 to 4 | 27.39 ± 0.153 | 59.94 ± 0.147 | 71.03 ± 0.215 | 59.43 ± 0.140 | 60.55 ± 0.125 |
| Kigezi (9) | Above 4 | 35.44 ± 0.131 | 64.06 ± 0.097 | 74.51 ± 0.107 | 61.38 ± 0.125 | 62.65 ± 0.121 |
| Mubende (198) | Above 4 | 37.65 ± 0.244 | 65.56 ± 0.156 | 77.69 ± 0.184 | 61.73 ± 0.116 | 62.86 ± 0.101 |
| Small East African (160) | Above 4 | 28.29 ± 0.154 | 60.02 ± 0.118 | 72.25 ± 0.180 | 60.11 ± 0.130 | 61.17 ± 0.118 |
| Significance | Breed | *** | *** | *** | *** | *** |
| | Age range | *** | *** | *** | *** | * |
| | Breed*Age range | * | ns | *** | ns | ns |

**Significance codes**: 0 '***' 0.001 '**' 0.01 '*' 0.05 '.' 0.1 ' ' 1

significant differences in body length, withers, and rump height, across breeds are independent of the age category and *vice versa*. Mubende and Kigezi goats have significantly (p≤0.0001) higher linear body characteristics compared to Small East African goats. Across age ranges, body length, height at withers, and height at rump significantly (p≤0.0001) increased from 2 to 3 years to >3 to 4 years, but were not significantly (p ≥ 0.05) different for goats of >3 to 4 and above 4 years.

## 4. Discussion

This study involved 1020 does from Mubende, Kigezi, and Small East African goat breeds, which were evaluated for body weight, linear body characteristics (body length, chest girth, withers height, rump height) and age across 10 agroecological zones in Uganda. FAO-recorded average body weights for Mubende and Kigezi goats are below the observed average body weight for the same breeds within the known home agroecological zones. Mubende goats from the western savannah grasslands zone weigh about 37kg and Kigezi goats in highland ranges weigh 34kg. The observed disparities in body weight of Mubende and Kigezi goats are probably a result of changes in animal management from mainly tethering with free-ranging only in non-cropping seasons in 1991 [6] to the current mainly free-range grazing system that allows greater access to forage than tethering. Free-range grazing improves growth rate through increased forage access time compared to tethering, which limits the diet quantity received per day by the animals [17].

For the first time, body weight and body size measurements of Uganda's goat breeds have been compared across agroecological zones, contributing important additional information to the national goat database. In general, all goat breeds had varying body weights and linear body measurements across agroecological zones. The observed variations could likely be an effect of genetic potential coupled with management and environmental conditions [18].

Mubende goats exhibited the highest body weight and linear body characteristics in their home zone, western savannah grasslands, which were comparable to those of Mubende goats in pastoral rangelands. The lowest body weight and linear measurements were observed for Mubende goats in the northwestern savannah grasslands and Lake Victoria crescent. The high body length, chest girth, withers height, and rump height for Mubende goats in western savannah grasslands probably result from the feeding system of free-range grazing and access to maize stover in the dry season, which allows all-year-round feed access. Many farmers in western savannah grasslands allowed goats to access maize stovers in fields after the cropping season for continued feed access when the grazing areas dried out. A combination of the fibrous maize stover and the proteinous browse tree species present in the savannahs allows goats to achieve the daily nutritional requirements in dry seasons [19], hence continued growth and/or body size increase. Mubende goats in pastoral rangelands showed higher body size characteristics relative to Mubende goats of comparable age in other agroecological zones, probably because the mosaic savannah vegetation in pastoral rangelands zone with a diversity of shrub to browse [3], coupled with free grazing management system that allows unrestricted access to a good quantity of forage for the goats to meet their nutritional requirements for body growth and maintenance [17]. Although the mean body weights of Mubende goats in other agroecological zones are below that observed in their home zone, they are comparable to the known average weight of 31kg [6], implying the breed is probably adapted to the different environments. However, the results of body weight and linear body measurements obtained in this study are above those observed by [20] for Mubende goats, probably because younger goats were included which had lower body size measurements thus reducing the overall average size for the breed. Variations in the skeletal size of goats may result from natural selection and

adaptation of the goat breed to a given environment where the genetic potential, hormones, and nutrient supply interact [12].

Kigezi goats in their home zone of highland ranges had the highest body size characteristics, which were comparable to those of Kigezi goats in western savannah grasslands and pastoral rangelands. The body size in highland ranges is indicative of the adaptation of Kigezi goats to the cool, moist mountainous environment in the home zone, as their long and curly hair coat helps them to keep warm, thus preventing loss of energy for temperature regulation. Long and coarse hair protects goats from heat loss in cold environments [21]. However, free-range grazing with dry season access to maize stover in Western savannah grasslands provides extra energy towards the unmet nutritional requirements [18]. In addition, extensive grazing on the mosaic savannah in the pastoral rangelands with a mixture of tree forages and shrubs also allows access to better quality and quantity of forage for the growth and maintenance requirements of the goats.

Small East African goats are known as small and hardy, with a body weight of 25 to 30kg [8]. A similar body weight trend was observed during this study for the Small East African goats in their home zone, northeastern drylands, which is probably an adaptation for thermotolerance [21]. However, Small East African goats in the home zone of northwestern savannah grasslands showed lower than average body weight with the lowest linear body characteristics compared to the same goats in other zones. Differences in body weight and body size of the Small East African goats in the known home zones of northeastern drylands and northwestern savannah grasslands may imply divergence in the breed characteristics due to differences in environmental characteristics and animal management between the two zones. Small East African goats in northeastern drylands freely graze and move long distances with the herders during the long, hot, and dry seasons in search of forage. Small East African goats in northeastern drylands are taller than those in other zones, probably for survival, given the hot and dry conditions. The long legs help goats prevent the ground radiation from affecting their body thus imparting them the potential to survive while grazing in the hot, arid, and semi-arid environments [22]. However, goats in northwestern savannah grasslands are mainly tethered and only freely graze outside the cropping seasons, which limits feed access and, hence, growth limitations. Similarly, results in [20] showed that Small East African goats from eastern Uganda had higher body size measurements as compared to the same breed in northwestern Uganda. However, the body sizes reported for the Small East African goats studied in [20] are lower than values reported by both FAO and this study. Variation in the skeletal size of goats is due to the effects of natural selection and adaptation of the various breeds or types of goats to different ecological regions.

In addition, the body size of all goat breeds increased with increasing age as a result of growth processes where body composition and conformation change with increasing age, resulting in increasing body dimensions [18]. The age and breed of a goat have an interactive influence on the body weight and chest girth of goats. It is also noted that the known average weight for all goat breeds based on FAO records was observed at 2 years, after which body weight increased for all goat breeds.

Generally, agroecological zones in Uganda influence body size characteristics of goats through the management systems implemented by the farmers as they cope with the various environmental characteristics. As goats adapt to different environments, morphological adjustments may occur to cope with the existing stressors, thus surviving, feeding, and reproducing under the prevailing conditions [23]. The variations in morphometric measurements relative to agroecological zones reported in the current study are similar to those earlier reported in Botswana among Tswana goats [24] and among indigenous female goats in Limpopo province in South Africa [25].

The results of this study clarify the phenotypic characteristics of indigenous goat breeds of Uganda, revealing their distribution, morphology, and body weight differences across agroecological zones. The information can be used as a basis for designing strategic management options among indigenous goats, including nutritional and breed improvement strategies. This knowledge is helpful in the planning and management of Uganda's indigenous goats for sustainability.

## 5. Conclusions

This study evaluated body weight and linear body size characteristics of indigenous Ugandan goat breeds across agroecological zones. Mubende and Kigezi goats showed improvement in body weight in their home zones relative to the known FAO values. Small east African goats in their home zone of northeastern drylands are physically larger compared to the same goats in the home zone of northwestern savannah grasslands. Across zones, Mubende goats also perform better in pastoral rangelands, while Kigezi goats are well adapted to western savannah grasslands and pastoral rangelands. Similarly, linear body size characteristics were high for each breed within the home zone than in other zones, and this information has added value to the existing phenotypic descriptions of indigenous goat breeds in Uganda. Understanding the current status of body weight and body size of Uganda's goat breeds across all agroecological zones contributes to the national and international data base for these goat breeds. Information from this study can be used to guide decision-making in selection of goat breeds suitable for an agroecological zone. Body weight and body size of a goat breed can be a basis for designing nutrition improvement studies and selection of suitable breeds for crossbreeding of the goats in different agroecological zones. A comparative understanding of breeds' performance across agroecological zones can assist farmers in setting up target body characteristics for marketing the different goat breeds at the farms. However, there is need to understand the genetic structure of Uganda's indigenous goats for precise exploitation of the unique factors underlying survival and productivity in the various environments.

## 6. Implications of the study

Differences in environmental characteristics need to be considered during breed choice so that a breed is raised in a zone where its production efficiency can be maximized.

The difference in performance of the goat breeds across agroecological zones show the importance of conservation of biodiversity of animal and environmental resources for a sustainable farming system.

## Supporting information

**S1 Data.**
(ZIP)

## Author Contributions

**Conceptualization:** Ziwena Nantongo, Morris Agaba, Swidiq Mugerwa, Josephine Birungi.

**Data curation:** Ziwena Nantongo, Morris Agaba.

**Formal analysis:** Ziwena Nantongo, Stephen Opiyo.

**Funding acquisition:** Josephine Birungi.

**Investigation:** Ziwena Nantongo.

**Methodology:** Ziwena Nantongo, Swidiq Mugerwa, Raphael Mrode, Josephine Birungi.

**Project administration:** Josephine Birungi.

**Resources:** Josephine Birungi.

**Software:** Ziwena Nantongo, Stephen Opiyo, Raphael Mrode.

**Supervision:** Morris Agaba, Gabriel Shirima, Swidiq Mugerwa, Raphael Mrode, Josephine Birungi, Linus Munishi.

**Validation:** Morris Agaba, Gabriel Shirima, Swidiq Mugerwa, Stephen Opiyo, Josephine Birungi, Linus Munishi.

**Visualization:** Linus Munishi.

**Writing – original draft:** Ziwena Nantongo.

**Writing – review & editing:** Morris Agaba, Gabriel Shirima, Swidiq Mugerwa, Stephen Opiyo, Raphael Mrode, Josephine Birungi, Linus Munishi.

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
