## [Decision Letter · Decision Letter 0]

3 Oct 2023

PONE-D-23-28075Variability in body weight and morphology of Uganda’s indigenous goat breeds across agro-ecological zonesPLOS ONE

Dear Dr. Nantongo,

Thank you for submitting your manuscript to PLOS ONE. After careful consideration, we feel that it has merit but does not fully meet PLOS ONE’s publication criteria as it currently stands. Therefore, we invite you to submit a revised version of the manuscript that addresses the points raised during the review process.

ACADEMIC EDITOR: The authors should provide current FAO statistics on the population of goats in the study area.Effect of sex and its interaction was not considered in the model despite sampling different sexes. There should be a concrete justification for this.There is a need to subject the manuscript to grammar checkThe queries of the two reviewers should also be adequately addressed.==============================

We look forward to receiving your revised manuscript.

Kind regards,

Abdulmojeed Yakubu

Academic Editor

PLOS ONE

“The research was funded by Swedish International Development Cooperation Agency (SIDA) through a grant to Biosciences eastern and central Africa—International Livestock Research Institute (BecA-ILRI Hub) (Sida contribution no: 51050080) to Nantongo Ziwena as a PhD fellow under the Africa Biosciences Challenge Fund (ABCF) program. The ABCF Pro-gram is funded by the Australian Department for Foreign Affairs and Trade (DFAT) through the BecA-CSIRO partnership, the Syngenta Foundation for Sustainable Agriculture (SFSA), the Bill & Melinda Gates Foundation (BMGF), the UK Department for International Development (DFID), and the Swedish International Development Cooperation Agency (SIDA).”

6. We note that Figures 1, 2, 3 and 5 in your submission contain copyrighted images. All PLOS content is published under the Creative Commons Attribution License (CC BY 4.0), which means that the manuscript, images, and Supporting Information files will be freely available online, and any third party is permitted to access, download, copy, distribute, and use these materials in any way, even commercially, with proper attribution. For more information, see our copyright guidelines: http://journals.plos.org/plosone/s/licenses-and-copyright.

1. You may seek permission from the original copyright holder of Figures 1, 2, 3 and 5 to publish the content specifically under the CC BY 4.0 license.

7. We note that Figure 4 in your submission contain [map/satellite] images which may be copyrighted. All PLOS content is published under the Creative Commons Attribution License (CC BY 4.0), which means that the manuscript, images, and Supporting Information files will be freely available online, and any third party is permitted to access, download, copy, distribute, and use these materials in any way, even commercially, with proper attribution. For these reasons, we cannot publish previously copyrighted maps or satellite images created using proprietary data, such as Google software (Google Maps, Street View, and Earth). For more information, see our copyright guidelines: http://journals.plos.org/plosone/s/licenses-and-copyright.

1. You may seek permission from the original copyright holder of Figure 4 to publish the content specifically under the CC BY 4.0 license. 

Reviewers' comments:

Reviewer's Responses to Questions

**Comments to the Author**

1. Is the manuscript technically sound, and do the data support the conclusions?

Reviewer #1: Partly

Reviewer #2: Yes

2. Has the statistical analysis been performed appropriately and rigorously? 

Reviewer #1: No

Reviewer #2: Yes

3. Have the authors made all data underlying the findings in their manuscript fully available?

Reviewer #1: Yes

Reviewer #2: Yes

4. Is the manuscript presented in an intelligible fashion and written in standard English?

Reviewer #1: Yes

Reviewer #2: Yes

5. Review Comments to the Author

Reviewer #1: he study was conducted in 10 different agroecological regions of Uganda to investigate the impact of environmental factors on the physical characteristics and live weight of domestic goat breeds. However, in the study, only the ages of the animals included in the experiment and the regions where they were raised were considered as environmental factors. Therefore, the purpose of the study should be aligned with the implemented applications. In this study, we conducted a basic identification study on a total of 1020 animals from 3 different breeds bred in Uganda. Considering the analyses carried out, it was deemed a significant shortcoming that the live weights of these breeds were not included as a covariate in the mathematical model used. On the other hand, the absence of specific statistics (such as mean, standard deviation, coefficient of variation, etc.) for races in the study makes it difficult to comprehend the level of variation. However, based on the values depicted in the charts presented in the article, it is evident that there is a significant variation among the breeds that were studied. It is important that the number of samples is not provided in the presented tables so that the reader can have a clearer understanding of the subject. The information provided in the conclusion of the article is purely informative. However, this section does not clearly state how the findings will be used as a foundation for future studies or how they will contribute to the current situation. Considering all the issues I mentioned above, the presented study should be revised statistically, and the conclusion section should be reorganized. The article was not considered worthy of publication in its current form for several reasons.

Reviewer #2: The paper discusses on data of morphometrics of different breeds of goats. The data is worthy of publication.

However, the presentation could be improved. Suggestions:

1. Figures 1, 2, and 3 . The pictures of the goat could be taken with not at an angle but on a straight view with roughly the same scale.. The pictures should be in Figures 1, 2 and 3 could be put on 1 page as Figures 1(a) (b) (c), in order to make identifying the goat breeds easier. No cropped image of the goat. The goat should be in full. Please improve the image of Fig 1 and Fig 2.

By putting the images side by side, the reader should be able to compare the goats visually.

2. Figure 4: The map and also the legend needs improving. (Blurred).

3. Figure 6: Where is the standard error /standard deviation for the graphs? Insert the SD/SE bars and add it to the caption of the Graph. Delete the word "A graph of"

4.Table 1 is cropped. Please improve.

5. Explain more about Table 4. and 3.3 Body size characteristics of goat breeds across age ranges, since there is significance among the breeds, and age range.

6. Is this the first report on Mubende, Kigezi, and Small East African goat breeds? Compare your data with other morphometric data of previous studies.

7. Please check the reference format.

6. PLOS authors have the option to publish the peer review history of their article (what does this mean?). If published, this will include your full peer review and any attached files.

Reviewer #1: **Yes: **Assoc. Prof. Dr. Onur YILMAZ

Reviewer #2: No

---

## [Author Response · Author response to Decision Letter 0]

14 Nov 2023

Dear Reviewers,

I appreciate the comments and insightful guidance given to my work. I have been able to respond to each comment by positively working on the changes to improve the manuscript. The comments' responses are as below and the same have been attached in the Rebuttal letter.

Response to reviewers’ comments

Academic editor’s comments

Academic editor’s comment Response

1. The authors should provide current FAO statistics on the population of goats in the study area. 

The population of goats in Uganda (10.4 million) as per current FAO statistics has been provided

2. Effect of sex and its interaction was not considered in the model despite sampling different sexes. There should be a concrete justification for this. This study sampled only female goats for the purpose of further genetic characterization which was based on mitochondrial DNA. Thus, the effect of sex was not applicable

3. There is a need to subject the manuscript to grammar check The manuscript has been subjected to grammar check using Grammarly 

The queries of the two reviewers should also be adequately addressed. The queries of the two reviewers have been addressed

Reviewer 1 comments

1. The study was conducted in 10 different agroecological regions of Uganda to investigate the impact of environmental factors on the physical characteristics and live weight of domestic goat breeds. However, in the study, only the ages of the animals included in the experiment and the regions where they were raised were considered as environmental factors. Therefore, the purpose of the study should be aligned with the implemented applications. The purpose of the study has been changed to ‘effect of agroecological zone on physical characteristics and body weight of Uganda’s indigenous goats’

 to match the implemented applications

2. In this study, we conducted a basic identification study on a total of 1020 animals from 3 different breeds bred in Uganda. Considering the analyses carried out, it was deemed a significant shortcoming that the live weights of these breeds were not included as a covariate in the mathematical model used. On the other hand, the absence of specific statistics (such as mean, standard deviation, coefficient of variation, etc.) for races in the study makes it difficult to comprehend the level of variation. However, based on the values depicted in the charts presented in the article, it is evident that there is a significant variation among the breeds that were studied. It is important that the number of samples is not provided in the presented tables so that the reader can have a clearer understanding of the subject. From the view point of the data presented, the only variable that could influence the outcome of other variables (Covariate) is age of the animals. Body weight of animals does not influence the body size measurements. Furthermore, variations in body weight was a factor of interest for this study, to understand the effect of agroecological zone on the body weight of goat breeds. Therefore, age of individual animals has been included in the ANOVA model as a covariate in the re-analysis which used ANCOVA procedures.

The numbers of samples used at the different levels of analysis have been added to the tables of results

3.The information provided in the conclusion of the article is purely informative. However, this section does not clearly state how the findings will be used as a foundation for future studies or how they will contribute to the current situation. More information about how the study findings can be used for further studies has been added to the conclusion

4. Considering all the issues I mentioned above, the presented study should be revised statistically, and the conclusion section should be reorganized. All issues mentioned have been considered and worked upon.

Reviewer 2 comments

1. Figures 1, 2, and 3. The pictures of the goat could be taken with not at an angle but on a straight view with roughly the same scale. The pictures should be in Figures 1, 2 and 3 could be put on 1 page as Figures 1(a) (b) (c), in order to make identifying the goat breeds easier. No cropped image of the goat. The goat should be in full. Please improve the image of Fig 1 and Fig 2.

By putting the images side by side, the reader should be able to compare the goats visually. 

The goat pictures have been replaced with those taken at relatively less angle position and all figures have been put to one page as Figure 1 (A) (B) (C). there is no cropped image among the new photos

2. Figure 4: The map and also the legend needs improving. (Blurred). The map and legend have been improved

3. Figure 6: Where is the standard error /standard deviation for the graphs? Insert the SD/SE bars and add it to the caption of the Graph. Delete the word "A graph of" Standard error bars have been included on the graph and caption added to the title. The line “A graph of” has been deleted

4. Table 1 is cropped. Please improve. Table 1 with all the rows and columns has been added

5. Explain more about Table 4. and 3.3 Body size characteristics of goat breeds across age ranges, since there is significance among the breeds, and age range. More explanation for the effects of breed and age category has been given (lines 312-323)

6. Is this the first report on Mubende, Kigezi, and Small East African goat breeds? Compare your data with other morphometric data from previous studies. This is the first study to compare morphometric data about Mubende, Kigezi, and Small East African goat breeds across agroecological zones. However, one study on overall body size variability for Mubende (Lines 363 - 366)and Small East African goats (lines: 396 – 399) has been used 

7. Please check the reference format. Format of references checked and corrected

---

## [Decision Letter · Decision Letter 1]

11 Dec 2023

Variability in body weight and morphology of Uganda’s indigenous goat breeds across agro-ecological zones

PONE-D-23-28075R1

Dear Dr. Nantongo,

We’re pleased to inform you that your manuscript has been judged scientifically suitable for publication and will be formally accepted for publication once it meets all outstanding technical requirements.

Kind regards,

Abdulmojeed Yakubu

Academic Editor

PLOS ONE

Additional Editor Comments (optional):

Reviewers' comments:

Reviewer's Responses to Questions

**Comments to the Author**

1. If the authors have adequately addressed your comments raised in a previous round of review and you feel that this manuscript is now acceptable for publication, you may indicate that here to bypass the “Comments to the Author” section, enter your conflict of interest statement in the “Confidential to Editor” section, and submit your "Accept" recommendation.

Reviewer #1: All comments have been addressed

Reviewer #2: All comments have been addressed

2. Is the manuscript technically sound, and do the data support the conclusions?

Reviewer #1: Yes

Reviewer #2: Yes

3. Has the statistical analysis been performed appropriately and rigorously? 

Reviewer #1: Yes

Reviewer #2: Yes

4. Have the authors made all data underlying the findings in their manuscript fully available?

Reviewer #1: Yes

Reviewer #2: Yes

5. Is the manuscript presented in an intelligible fashion and written in standard English?

Reviewer #1: Yes

Reviewer #2: Yes

6. Review Comments to the Author

Reviewer #1: The authors made all the corrections requested within the scope of referee criticism. Therefore, it is appropriate to publish the article in its current form.

Reviewer #2: Good fundamental data on indigenous Ugandan goat breeds. The cross-sectional survey was conducted in 323 households 24 from the ten zones, where 1020 goats composed of three breeds (Mubende, Kigezi, and Small

25 East African) were sampled and measured for body weight, linear body size, and age.

7. PLOS authors have the option to publish the peer review history of their article (what does this mean?). If published, this will include your full peer review and any attached files.

Reviewer #1: **Yes: **ONUR YILMAZ

Reviewer #2: **Yes: **Dr Shahrizim Zulkifly

---

## [Editor Report · Acceptance letter]

19 Dec 2023

PONE-D-23-28075R1 

PLOS ONE

Dear Dr. Nantongo, 

I'm pleased to inform you that your manuscript has been deemed suitable for publication in PLOS ONE. Congratulations! Your manuscript is now being handed over to our production team.

Kind regards, 

on behalf of

Professor Abdulmojeed Yakubu 

Academic Editor

PLOS ONE